# Does Standing Up Enhance Performance on the Stroop Task in Healthy Young Adults? A Systematic Review and Meta-Analysis

**DOI:** 10.3390/ijerph20032319

**Published:** 2023-01-28

**Authors:** Maja Maša Šömen, Manca Peskar, Bettina Wollesen, Klaus Gramann, Uros Marusic

**Affiliations:** 1Department of Psychology, Faculty of Arts, University of Ljubljana, Aškerčeva Cesta 2, 1000 Ljubljana, Slovenia; 2Science and Research Centre Koper, Institute for Kinesiology Research, Garibaldijeva Ulica 1, 6000 Koper, Slovenia; 3Biological Psychology and Neuroergonomics, Department of Psychology and Ergonomics, Faculty V: Mechanical Engineering and Transport Systems, Technische Universität Berlin, Straße des 17. Juni 135, 10623 Berlin, Germany; 4Human Movement and Training Science, Institute of Human Movement Science, Psychology and Human Movement, University Hamburg, Turmweg 2, 20146 Hamburg, Germany; 5Department of Health Sciences, Alma Mater Europaea—ECM, Slovenska Ulica 17, 2000 Maribor, Slovenia

**Keywords:** healthy young adults, dual task, posture, Stroop task, cognitive-motor interference, sit-to-stand workstations

## Abstract

Understanding the changes in cognitive processing that accompany changes in posture can expand our understanding of embodied cognition and open new avenues for applications in (neuro)ergonomics. Recent studies have challenged the question of whether standing up alters cognitive performance. An electronic database search for randomized controlled trials was performed using Academic Search Complete, CINAHL Ultimate, MEDLINE, PubMed, and Web of Science following PRISMA guidelines, PICOS framework, and standard quality assessment criteria (SQAC). We pooled data from a total of 603 healthy young adults for incongruent and 578 for congruent stimuli and Stroop effect (mean age = 24 years). Using random-effects results, no difference was found between sitting and standing for the Stroop effect (Hedges’ *g* = 0.13, 95% CI = −0.04 to 0.29, *p* = 0.134), even when comparing congruent (Hedges’ *g* = 0.10; 95% CI: −0.132 to 0.339; *Z* = 0.86; *p* = 0.389) and incongruent (Hedges’ *g* = 0.18; 95% CI: −0.072 to 0.422; *Z* = 1.39; *p* = 0.164) stimuli separately. Importantly, these results imply that changing from a seated to a standing posture in healthy young adults is unlikely to have detrimental effects on selective attention and cognitive control. To gain a full understanding of this phenomenon, further research should examine this effect in a population of healthy older adults, as well as in a population with pathology.

## 1. Introduction

Everyday life often involves performing a postural task (single task, ST) such as sitting or standing concurrently with another (dual task, DT [1,2]). This area of research is confronted with conflicting results regarding observations that can be made when a sitting or standing task is combined with an additional cognitive task [3]. Most of the data on this topic come from studies that examined the effects of sit-to-stand or standing workstations on cognitive performance in the work environment. The recent shift from physically intensive jobs to computer-based jobs has reduced work-related physical activity and indirectly promoted prolonged sitting periods [4]. It has been postulated that reducing the proportion of time spent seated in the workplace with interventions such as sit-to-stand or standing workstations can reduce absenteeism and the risk of developing chronic health problems later in life [5,6], as well as increase work productivity and quality of life [7,8]. However, it remains unclear whether body position (i.e., sitting or standing) results in a change in cognitive function or work productivity [9]. Within this research tradition, two opposite observations were made: (1) standing compared to sitting with an additional cognitive task leads to performance decrements either of the cognitive or motor performance (cognitive-motor interference), and (2) standing compared to sitting with an additional cognitive task leads to performance benefits, especially for the motor task.

Some studies suggest that standing impairs cognitive performance. For example, Roerdink et al. [10] inferred that cognitive resources invested in standing postures were greater than those invested in sitting. Despite relatively small effect sizes on average, Schraefel et al. [11] found significantly better performance on relatively complex cognitive tasks involving attention and executive functions (e.g., ability to maintain a sustained focus, resist distraction, switch attention between tasks, and process information) when sitting compared to standing. Similarly, Kang et al. [12] found significant decreases in cognitive performance (degraded attention and executive function) when engaged in a task at a standing workstation compared to a sitting workstation, an effect even more pronounced in a cognitive task with a higher difficulty level.

Conversely, standing is associated with better cognitive performance than sitting. It appears that individuals in a standing position show a greater tendency to engage in controlled cognitive processing than individuals who are seated. For example, Smith et al. [13] found reduced Stroop interference (i.e., the difference between incongruent and neutral trials), slower search rates, and reduced switch costs in the visual search task when standing compared to sitting. In addition, Liu and Liao [14] found that participants in a standing posture agreed more with the utilitarian proposal and became less deontological.

There are, however, also studies that show no significant difference in cognitive performance between the two postures. Russell et al. [15] found no significant difference between sitting and standing in processing speed, working memory, or attention. Similarly, Schwartz et al. [16] found no change in working speed or attention in the standing compared to the seated position. Furthermore, no difference in visual, verbal, or auditory reaction time [17] and no dual-task cost (i.e., the percentage of decrements in the performance of a dual task relative to the performance of a single task) has been observed between standing and sitting [18]. Lastly, Rostami et al. [19] found no statistically significant difference between cognitive and skill performance between the two postures.

The most commonly used cognitive tasks that assess inhibitory control are the Stroop task, antisaccade tasks (e.g., *n*-back task, 3-back task, and 2-back task), Simon task, Flanker task, and a Go/NoGo task [20]. The Stroop task requires participants to respond to the color in which a letter string appears while ignoring the meaning of the string (e.g., the word “BLUE” presented in red ink). Although there are many variations of the Stroop task [21], the standard Stroop paradigm consists of congruent and incongruent stimuli. Congruent stimuli are those in which the letter string spells out the name of the same color in which the string is written (e.g., the word “RED” presented in red ink), whereas incongruent stimuli involve the letter string that spells out the name of an alternative color to the one in which the string is written (e.g., the word “BLUE” presented in red ink) [13]. With incongruent stimuli, participants generally take more time to respond and make more errors (commonly referred to as the Stroop effect) because of the interference that occurs between the meaning of the letter string and the actual color in which the string is written [22]. The *n*-back task (e.g., 3-back task, 2-back task) is a widely used measure of working memory, where participants monitor a series of stimuli and are required to respond whenever the same stimulus is presented as it was presented in *n* trials (e.g., 3 or 2) previously [23,24]. The Simon task is a two-choice reaction-time task where a stimulus is presented on the left or right side of the screen and participants are instructed to select the response based on a stimulus content while ignoring the stimulus location. Similar to the Stroop effect, the Simon effect is the finding that participants are generally faster when the irrelevant stimulus location and the response key location correspond [25]. The Flanker task is a measure of selective attention where participants are instructed to ignore irrelevant congruent flanker stimuli (i.e., associated with the same response as the target) or incongruent flanker stimuli (i.e., associated with the opposite response as the target) while they categorize a target stimulus as fast and accurately as possible. Participants generally take longer and are less accurate with incongruent flanker stimuli, which is called the Flanker congruency effect [26]. Finally, in a Go/NoGo task, participants are instructed to perform a motor response when certain stimuli (i.e., targets) are displayed on the screen or to withhold this motor response for other stimuli (i.e., non-targets) [27].

Despite several above-mentioned inhibitory control tasks, the Stroop task has recently gained a lot of interest concerning differences in cognitive performance during sitting and standing. In their study with young (undergraduate) students, Rosenbaum et al. [28] found that the Stroop effect was smaller while completing the task in the standing position than in the sitting position, suggesting that standing reduces the magnitude of the Stroop effect because the added requirement of controlling posture while standing, as opposed to sitting, entails extra attentional load and stress, which in turn improves the selectivity of attention. Similarly, Smith et al. [13] also found an influence of posture on the magnitude of the Stroop effect in undergraduate students, whereas Caron et al. [29] found no interaction. Regardless of the presence of neutral trials, mode of response (vocal or manual), and whether participants stood on one or two feet, posture did not affect the magnitude of the Stroop effect. Caron et al. [29] ascribed their failure to replicate the findings of Rosenbaum et al. [28] and Smith et al. [13] either to a subtle factor (e.g., important differences in participant populations across institutions) or to the unreliability of the influence of posture on the magnitude of the Stroop effect. Nevertheless, they suggested that the finding that postural differences between sitting and standing do not influence the magnitude of the Stroop effect implies that the use of sit-to-stand workstations will likely not have a deleterious impact on cognitive performance. Recently, Straub et al. [30] published two conceptual replications of the studies by Rosenbaum et al. [28] and a meta-analysis on the question of whether body posture reduces the Stroop effect, finding no influence of posture on the Stroop effect. Moreover, using Bayesian analysis, they found strong evidence against body posture and the Stroop effect. Their meta-analytic findings did not support the claim that standing modulates the Stroop effect. Given that there are currently several recent reports addressing the Stroop task in both sitting and standing, reporting unchanged or facilitated cognitive processing in standing, this systematic review and meta-analysis aimed to investigate how sitting compared to standing affects performance in the Stroop task. Following the study by Straub et al. [30], we included three additional studies in our meta-analysis and evaluated them separately not only for the Stroop effect but also for congruent and incongruent stimuli.

## 2. Methods

This review complies with the preferred reporting items for systematic reviews and meta-analyses (PRISMA) guidelines [31]. Methods for obtaining and handling data and inclusion criteria were specified in advance and registered with the International Prospective Register of Systematic Reviews (PROSPERO ID: CRD42021242451). Furthermore, the problem/population, intervention, comparison, outcome, and study (PICOS) design framework [32,33,34] was followed for the literature search strategy and selection criteria:Population: healthy adultsIntervention: word-color Stroop taskComparison: sitting vs. standing postureOutcome measures: Stroop task performance (reaction times)Study design: crossover randomized controlled trials

### 2.1. Search Strategy

The databases Academic Search Complete, CINAHL Ultimate, MEDLINE, PubMed, and Web of Science were searched to obtain relevant empirical articles written in English. The search strings applied were identical in all five search tools and were intended to capture all articles on how standing vs. sitting affects Stroop task performance, specifically reaction times. Articles published by 31 December 2022 were considered, and specific deviations of keyword combinations comprising “stroop”, “posture”, “sit”, “stand”, “work desk”, and “workstation” were used in the identification process (a detailed list of key terms is added in Table A1). The search results were managed in Mendeley with duplicates removed.

### 2.2. Selection Criteria

The study selection was limited to crossover randomized controlled trials that met the following criteria: (a) the article was written in English; (b) the basis of the article was empirical; (c) participants were 18–65 years old; (d) data from healthy/normative samples without any cognitive deficits were reported; (e) the Stroop test was performed; (f) research paradigm included both sitting and standing condition; and (g) reaction times were presented or potentially available. Two reviewers (M.M.Š. and M.P.) independently reviewed the titles and abstracts to identify all potentially eligible articles following the PRISMA [31] and PICOS [32,33,34] methodologies. Afterward, the two reviewers independently assessed full-version copies of all potentially eligible articles to determine the ones to be included, with additional screening of the reference list of each included article. Any disagreement on inclusion was resolved by discussion and arbitration by three reviewers (U.M., K.G., and B.W.). The set of standard quality assessment criteria (SQAC) for evaluating primary research papers was used to assess the methodological quality of the studies included in this review [35]. Participant selection (c) was verified by comparing the sample with the conclusions drawn from the experimental results. When the sample consisted of undergraduate students, but results were generalized across healthy adults of different ages, a general remark was noted. A full point for a sufficient description of patient characteristics (d) was given when the gender proportion was mentioned and when participants’ health conditions were assessed and described (e.g., the absence of cognitive and motor deficits or injuries, or the absence of vision impairments). Only random allocation was assessed (e) as no interventions were carried out in the present studies. A full point was given when the recording of reaction times was well-defined in terms of when and how it was done (h). Appropriate sample size (i) evaluation was based on an exemplary calculation using G * Power software (Heinrich Heine University, Dusseldorf, Germany, Version: 3.1.9.2). The repeated-measures analysis of variance used in all 12 experiments required a sample size of at least 36 participants (critical *F* = 4.13) when an effect size *f* = 0.5 was assumed (α error: 0.05; power: 0.95), provided the study included at least two groups with four measurements. A full point for appropriate sample size was given when either an a priori calculation of sample size had been described or the sample size was at least 36 (for analysis of variance). Based on the report of the estimate of variance in main results (k), the standard deviation for congruent and incongruent stimuli had to be reported. A sufficiently detailed report of results (m) evaluation was based on the report of the Stroop effect in terms of average and standard deviation. MMŠ and MP performed the assessment independently and the results presented in Table 1 were concurred on. Each criterion, when complied with, was given one point. Points were added up and resulted in the quality score (range 0–12 points). The necessary score for a study of high quality was defined to be ≥10 out of 12 and ≥6–9 for standard quality according to the SQAC. No point was given if general remarks had to be made (indicated by brackets).

### 2.3. Data Extraction and Statistical Analysis

The information obtained from each of the seven studies (Rosenbaum et al. (2017) [28]; Schwartz et al. (2018) [16]; Zhang et al. (2018) [36]; Smith et al. (2018) [13]; Caron et al. (2020) [29]; Straub et al. (2022) [30]; Pinho et al. (2022) [37]; for information on how studies were derived, see “Study Selection” section) included the first author’s name, the year of publication, study aims, study population (sample size, age, and sex), measures used and study design, exposure duration, outcome, and findings (see Table 2). The average, standard deviation, and sample size for reaction time for both sitting and standing posture were recorded for the Stroop effect. Furthermore, the average, standard deviation, and sample size for reaction time for both sitting and standing posture were separately recorded for congruent and incongruent stimuli. Where studies did not include adequate data on statistics, corresponding authors were contacted to request additional data, and/or missing data were calculated from publicly available raw data and pipeline routines. Analyses were performed using the program Comprehensive Meta-Analysis (CMA version 2, Biostat, Englewood, NJ, USA). Effect size (ES) estimates (Hedges’ *g*) were calculated for each eligible study using available values (*M* ± *SD*, *N*). Hedges’ *g* was calculated by dividing the difference between the sitting and standing means by the pooled and weighted standard deviation [38]:$${\mathrm{Hedges}}^{\prime }g=\frac{M\mathrm{sitting}-M\mathrm{standing}}{SD^{*}\mathrm{pooled}}$$

The standardized mean difference (Hedges’ *g*) for all trials was aggregated and interpreted as follows: small = 0.20, medium = 0.50, large = 0.8028. Fixed- and random-effects models were calculated to compare the robustness of the analyses; however, due to the heterogeneity of the study designs used (see Table 2), the random-effects model was used to interpret the results of this meta-analysis. The standardized weighted mean difference and its 95% confidence interval (CI) were calculated. Statistical heterogeneity was assessed using the *Q* and *I*^2^ statistics. The *I*^2^ measure of inconsistency was calculated to determine the degree of statistical heterogeneity: low (25%), moderate (50%), and high (75%) statistical heterogeneity [39]. In addition, due to the degree of heterogeneity between studies, a random-effects model was used for all comparisons. However, additional sensitivity analysis was conducted using both fixed and random effects, as well as by excluding each study from the model. Publication bias was assessed by the asymmetry of the funnel plot using Egger’s test, and significant bias was considered to exist when *p* < 0.10. For the meta-analyses, the significance level was set at *p* ≤ 0.05.

## 3. Results

### 3.1. Study Selection

A total of 736 articles were originally identified in Academic Search Complete, 435 in CINAHL Ultimate, 1283 in MEDLINE, 930 in PubMed, and 1354 in Web of Science. The initial search was subsequently reduced to 2261 after duplicate publications were removed. Upon reviewing the title and abstract, 335 potential publications were identified and underwent a full article review. In total, seven met the inclusion criteria, and 328 were excluded for the following reasons: age (18), study design (239), did not include a Stroop task (71), language (1), and unsuccessful correspondence with the author(s)/missing data (6). In three of the seven included studies, two (Straub et al., 2022 [30]), three (Rosenbaum et al., 2017 [28]), and five (Caron et al., 2020 [29]) experiments were performed, respectively, but only two experiments in the study of Rosenbaum et al. (2017) [28] and four experiments in the study of Caron et al. (2020) [29] fit the scope of the present work and were included in the meta-analysis. In the study of Pinho et al. (2022) [37], only incongruent stimuli were included. Therefore, the meta-analysis was performed on a total of 11 experiments for congruent stimuli and the Stroop effect, and 12 experiments for incongruent stimuli. Additional data were requested from 12 authors, 6 of whom provided feedback. Details of the study selection process are presented in Figure 1.

### 3.2. Characteristics of Included Studies

Overall, the 12 experiments (Rosenbaum et al. (2017)—EX1, EX3 [28]; Schwartz et al. (2018) [16]; Zhang et al. (2018) [36]; Smith et al. (2018) [13]; Caron et al. (2020)—EX1, EX2, EX3, EX5 [29]; Straub et al. (2022)—EX1, EX2 [30]; Pinho et al. (2022) [37]) included 603 participants for incongruent and 578 participants for congruent stimuli and Stroop effect, of which all participated in both sitting and standing conditions. There was a lot of missing data regarding participants’ characteristics, which were noted in the process of quality assessment (see Table 1). Participants’ mean chronological age was 24.25 (two studies only provided data about the participants’ level of studies being undergraduate students). Data about participants’ sex were provided only by three studies with approximately half of their participants being women, whereas one study included only women. One study was conducted in the USA (Iowa) and Canada (Ontario) [29], one in the USA (Washington) [13], one in South America (Brazil) [37], two in Asia (Israel [28]; China [36]), and two in Europe (Austria [16]; Germany [30]).

All 12 experiments included a color-word Stroop test with the number of stimuli presented ranging from 72 to 190 (*M* = 116,17). The majority of experiments consisted of half congruent and half incongruent stimuli (7/12), whereas four experiments also included a third of neutral stimuli and one only included incongruent stimuli. In six experiments, participants responded by speaking the name of the print color into a microphone, in five experiments participants were required to indicate the answer by pressing the corresponding button on the keyboard, and one experiment did not report the method used to obtain answers from participants. The protocol used in the two experiments included also other measures, such as the text editing task, d2R test, N-back test, and more-odd shifting task (see Table 2).

In nine experiments, participants performed the Stroop task in both sitting and standing conditions, and the order of testing between sitting and standing was counterbalanced randomly across participants. In one experiment, participants performed the Stroop task in sitting, standing, walking at an active workstation with self-selected speed (self-paced walking), and walking at an active workstation with 1.5 times the self-selected speed (faster walking) condition on four different occasions. Moreover, one experiment divided participants into the intervention arm where the Stroop task was executed in alternating postures (sit-stand-sit-stand-sit), and in the control arm (non-intervention day) where the Stroop task was conducted in a sitting posture (sit-sit-sit-sit-sit). Finally, in one experiment participants performed the Stroop task in a seated condition, standing, and standing with haptic input via light touch. These three experiments also included counterbalanced conditions. Only two experiments provided data about the duration of the exposure to the Stroop task (see Table 2).

### 3.3. Meta-Analysis Outcomes

For the Stroop effect, there was a small but nonsignificant effect (Hedges’ *g* = 0.13; 95% CI: 0.039 to 0.290; *Z* = 1.50; *p* = 0.134; see Figure 2). The heterogeneity across these findings was moderate (*Q* = 12.17; *I*^2^ = 34%; *p* = 0.144). Inspection of the funnel plot did not show asymmetry (Egger’s intercept = 1.84; *p* = 0.122; see Figure A1). Additional analyses revealed that the magnitude of the effect derived from random and fixed methods did not vary. The magnitude of the effect assessed by the random method showed a small but nonsignificant effect (Hedges’ *g* = 0.13; 95% CI: 0.039 to 0.290; *Z* = 1.50; *p* = 0.134). Similarly, the fixed method showed a small but nonsignificant effect (Hedges’ *g* = 0.10; 95% CI: 0.023 to 0.226; *Z* = 1.60; *p* = 0.110). Sensitivity analysis revealed that the magnitude of the effect was not modified after each study was excluded (range of Hedges’ *g* = 0.08; 95% CI: -0.080 to 0.231; *Z* = 0.95; *p* = 0.342 to Hedges’ *g* = 0.15; 95% CI: −0.001 to 0.305; *Z* = 1.95; *p* = 0.051).

For congruent stimuli, there was a small but nonsignificant combined effect size (Hedges’ *g* = 0.10; 95% CI: −0.132 to 0.339; *Z* = 0.86; *p* = 0.389; see Figure 3). There was heterogeneity across these findings (*Q* = 36.26; *I*^2^ = 72%; *p* < 0.001). Inspection of the funnel plot revealed no asymmetry (Egger’s intercept = −2.21; *p* = 0.307; see Figure A2). Additional analyses revealed that the magnitude of the effect derived from random and fixed methods varied. The magnitude of effect assessed by the random method showed a small but nonsignificant effect (Hedges’ *g* = 0.10; 95% CI: −0.132 to 0.339; *Z* = 0.86; *p* = 0.389), whereas the fixed method showed a significant small effect size (Hedges’ *g* = 0.16; 95% CI: 0.037 to 0.274; *Z* = 2.57; *p* = 0.010). Sensitivity analysis revealed that the magnitude of the effect was not modified after each study was excluded (range of Hedges’ *g*: 0.03; 95% CI: −0.157 to 0.211; Z = 0.29; *p* = 0.773 to Hedges’ *g*: 0.15; 95% CI: −0.083 to 0.387; Z = 1.27; *p* = 0.204).

For incongruent stimuli, there was a medium but nonsignificant effect (Hedges’ *g* = 0.18; 95% CI: −0.072 to 0.422; *Z* = 1.39; *p* = 0.164; see Figure 4). There was heterogeneity across these findings (*Q* = 44.88; *I*^2^ = 76%; *p* < 0.001). Inspection of the funnel plot did not show asymmetry (Egger’s intercept = −1.56; *p* = 0.485; see Figure A3). Additional analyses revealed that the magnitude of the effect derived from random and fixed methods varied. The magnitude of effect assessed by the random method showed a medium but nonsignificant effect (Hedges’ *g* = 0.18; 95% CI: −0.072 to 0.422; *Z* = 1.39; *p* = 0.164), whereas the fixed method showed a significant medium effect size (Hedges’ *g* = 0.22; 95% CI: 0.098 to 0.332; *Z* = 3.60; *p* < 0.001). Sensitivity analysis revealed that the magnitude of the effect was not modified after each study was excluded (range of Hedges’ *g* = 0.09; 95% CI: -0.098 to 0.279; *Z* = 0.94; *p* = 0.347 to Hedges’ *g* = 0.22; 95% CI: −0.021 to 0.468; *Z* = 1.79; *p* = 0.074).

## 4. Discussion

In light of recent findings suggesting that performance on the Stroop task is unchanged or improved in standing compared with sitting, this systematic review and meta-analysis aimed to investigate whether standing compared with sitting affects the reaction times for the Stroop effect and also specifically for congruent and incongruent stimuli in healthy young adults. A similar meta-analysis was recently published by Straub et al. [30], but their only focus was on the Stroop effect and did not investigate whether standing compared to sitting affects the reaction time for congruent and incongruent stimuli separately. The present meta-analysis revealed no significant difference between sitting and standing in Stroop performance. Although not significant, the overall effects lean toward facilitated cognitive processing while standing compared to sitting, and can therefore be used in various areas of (neuro)ergonomics where the optimal relationship between reduction in sedentarism and work productivity is explored. The overall non-negative trend observed in the present study of 603 healthy young adult participants for incongruent and 578 for congruent stimuli and the Stroop effect suggests that working while standing compared with sitting is unlikely to have adverse effects on selective attention and cognitive control.

The findings of our meta-analysis are in agreement with previous scientific attempts that report no significant difference in cognitive performance between sitting and standing postures [15,16,17,18,19]. Standing upright presents a relatively simple task that does not necessarily involve higher structures of the central nervous system [40]. In other words, normal standing is an automatized task leading to a reduced need for shared central resources, which means that two (or more) tasks can be performed simultaneously with minimal or no interference [41]. Although nonsignificant, overall effects lean toward facilitated cognitive processing while standing compared to sitting. This suggests a more enhanced selective attention and controlled cognitive processing in standing compared to sitting, which could be because physically fit individuals such as undergraduate students exhibit better individual abilities and resources not associated with any changes in brain structure and function [42,43]. Furthermore, even though standing is considered a form of low-intensity exercise [15] that can increase non-exercise activity thermogenesis (NEAT) [44] but does not necessarily lead to improved cognitive performance, the postural control demands associated with standing may result in heightened arousal—recruiting additional cognitive resources that are used for the task at hand [13,45]. The observed results that lean toward facilitated cognitive processing while standing compared to sitting in our experiment were even bigger for incongruent stimuli and the Stroop effect. It appears that in the population of healthy young adults, standing-engaged mechanisms yielded a more effective selection of task-relevant information [13].

Historically, sit-to-stand workstations have been found to benefit physical health, particularly chronic diseases [46]. In contrast, the effects of sit-to-stand workstations on cognitive performance remain unclear to this day [16,36,47,48]. Although the study by Caron et al. [29] found no differences in the magnitude of the Stroop effect between sitting and standing and therefore failed to replicate the findings of Rosenbaum et al. [28] and Smith et al. [13], it suggests that the use of sit-to-stand workstations is unlikely to have adverse effects on cognitive performance. Further evidence was recently provided by Straub et al. [30], who found no influence of posture on the Stroop effect in two conceptual replications and no support for the claim that standing modulates the Stroop effect in a meta-analysis. As noted above, our findings suggest that transitioning a workforce to sit-to-stand workstations or simply introducing regular sitting breaks is unlikely to result in any short-term decrease in cognitive performance (and work productivity).

It is essential to emphasize that we have focused only on studies of healthy young adults who rarely deal with impaired cognitive processing caused by dysregulation or a reduction in attentional resources for tasks requiring executive control [49] or because of the prioritization of postural control or gait stability over the performance of a concurrent cognitive task to reduce falls and injuries [50]. Therefore, healthy young adults might not have considered the standing task to be challenging or dangerous enough for losing postural control, which would have, in turn, led to the relocation of cognitive resources. Nevertheless, according to some studies, both healthy younger and older subjects prioritize balance [51,52], while others show that balance is prioritized only when participants are explicitly instructed to assign greater priority to balance [53,54]. With the task prioritization model of walking, Yogev-Seligman et al. [55] postulate that prioritizing has two basic requirements. The first one is the postural reserve, which is the ability to perform a balancing task effectively, and the second one is hazard estimation, which is the ability to recognize potential dangers posed by the environment and/or individual limitations. They speculate that healthy individuals with a high postural reserve and hazard estimation can prioritize the cognitive task for an extended period without any adverse effects on posture. When task complexity is rising, attention focus shifts to balance, but it does not necessarily affect performance in any way. For example, when increasing complexity from single to dual tasks, Shumway-Cook et al. [50] found increasing decrements in postural control but not in cognitive performance. Similarly, in their study of the influence of a visual–verbal Stroop test on the standing and walking performance of older adults, Wollesen et al. [56] found that the sway velocity and the depending sway length did increase significantly under dual-task conditions, whereas no significant decrements in cognitive performance could be observed during standing. Furthermore, one of the studies in this meta-analysis performed the Stroop task in sitting, standing, and walking at an active workstation with self-paced and faster walking and found that for both congruent and incongruent trials there was no significant difference in accuracy and response time across the four conditions [36].

Potential limitations of this meta-analysis need to be considered. First, participants’ characteristics were poorly described in all included studies and all participants were students, who might be familiar with the Stroop task. Yogev-Seligmann et al. [55] found that familiarity with the task also determines the number of attentional resources, leading to lower cognitive-motor interference. Moreover, because these effects may be more pronounced in an older age group, further research should examine the effects of standing versus sitting on selective attention and cognitive control in both a population of healthy older adults and a population with pathology. Second, we were only interested in how standing compared to sitting affects the Stroop effect in terms of reaction times. Further research should focus on the comparison of the three postural conditions, i.e., sitting, standing, and walking, as cognitive-motor interference rises with increasing task complexity of either motor or cognitive task [43,56]. It is also unclear whether the results would generalize over longer standing durations. It would also be interesting to conduct a similar meta-analysis with tasks measuring other executive functions, not only selective attention and cognitive control using the Stroop task. In addition, the clinical significance of standing is currently limited. Despite the known health benefits of physical activity, older adults generally remain sedentary for an average of 9.4 h per day (up to 80% of their waking day) [57]. These numbers are even higher for frail populations such as diabetics and those with Parkinson’s disease, suggesting that the need for physical activity is not being met among older adults who need it most. Considering the known barriers to physical activity participation in older patients with chronic conditions, reducing physical inactivity through intermittent non-exercise physical activity would be a logical first step. This raises the question of neurocognitive performance, which is currently poorly studied in such populations. Finally, to improve the validity of this study, smaller and portable technologies such as electroencephalography (EEG) or functional near-infrared spectroscopy (fNIRS) should be used to gain a more ecological and thorough insight into the processes behind cognitive-motor interference in healthy young adults. In addition, by combining the mobile brain imaging of participants during movement with synchronized recordings of task performance and body movements, mobile brain/body imaging (MoBI) [58,59,60] allows for a deeper insight into the brain dynamics underlying the cognitive-motor interaction during different dual-task conditions [61].

## 5. Conclusions

In conclusion, no significant difference was found between sitting and standing for the Stroop effect, even when comparing congruent and incongruent stimuli separately. However, there appears to be a nonsignificant trend toward facilitated cognitive processing in standing compared with sitting, particularly for the incongruent stimuli and the Stroop effect. Although this meta-analysis included only seven heterogeneous studies and should therefore be interpreted with caution, our results offer a better understanding of embodied cognition and can be used in various areas of (neuro)ergonomics. For example, in line with our results, the implementation of sit-to-stand workstations should not lead to a decrease in cognitive performance, specifically selective attention and cognitive control. Future research is needed to support these conclusions and should focus on assessing cognitive-motor interference in healthy older adults and pathological populations, as well as in more complex postural conditions and other executive functions. Nevertheless, caution should be exercised in interpreting these results, as the robustness of the analysis and generalization is limited.

## Figures and Tables

**Figure 1 ijerph-20-02319-f001:**
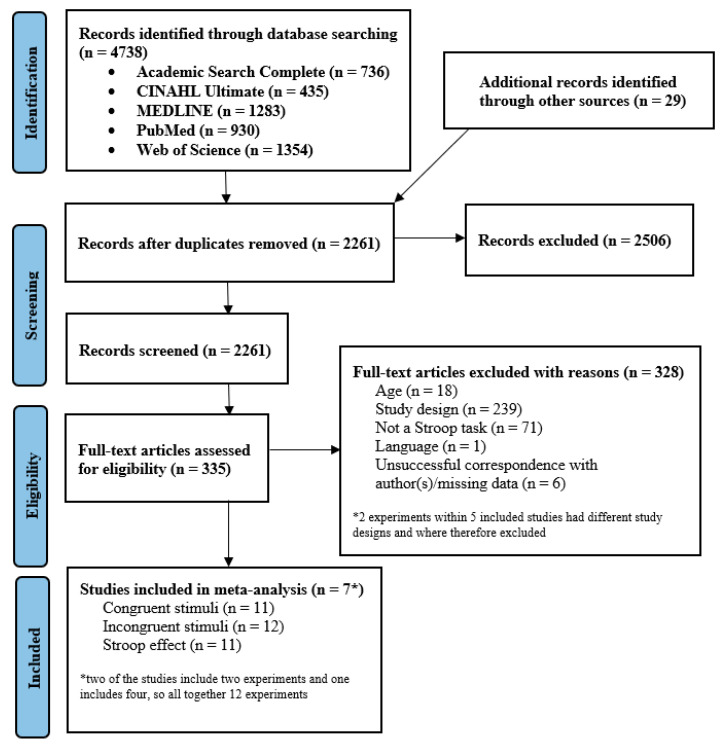
Flow diagram of article screening and selection process.

**Figure 2 ijerph-20-02319-f002:**
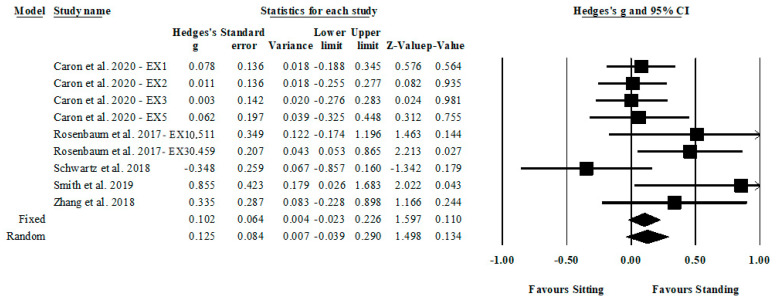
Forest plot of the Hedges’ *g* differences between sitting and standing conditions for the Stroop effect.

**Figure 3 ijerph-20-02319-f003:**
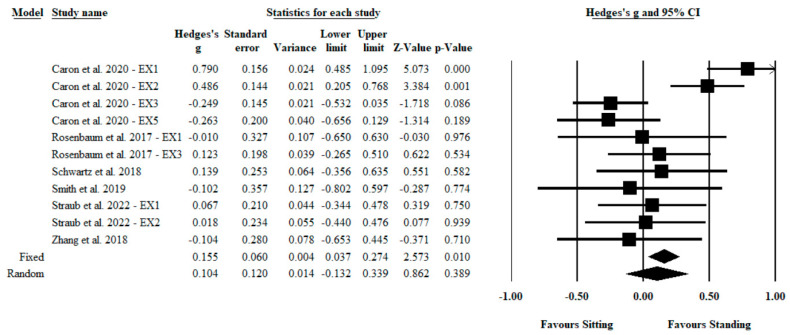
Forest plot of the Hedges’ *g* differences between sitting and standing conditions for congruent stimuli.

**Figure 4 ijerph-20-02319-f004:**
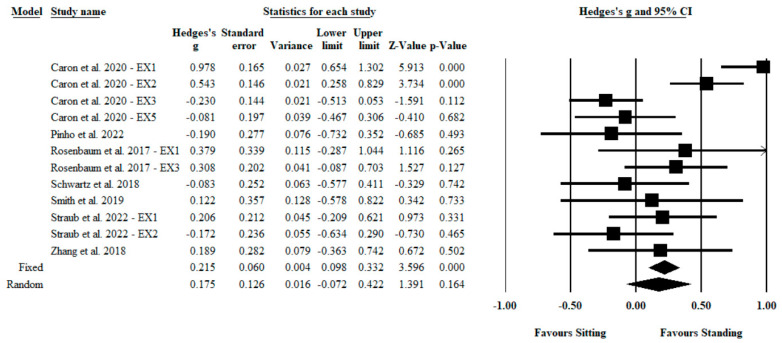
Forest plot of the Hedges’ *g* differences between sitting and standing conditions for incongruent stimuli.

**Table 1 ijerph-20-02319-t001:** Quality scores of the included studies and remarks.

Study	Experiment	Quality Criteria												Quality Score	Remark
		a	b	c	d	e	h	i	j	k	l	m	n		
Rosenbaum et al. (2017) [28]	EX1	x	x	(x)	(x)	x	(x)	x	x	x	x	(x)	x	8	c: undergraduate students; d: no information regarding participants’ gender and health condition (only vision was reported); h: no description of when and how the RTs were extracted specifically; m: no report of the Stroop effect (*M* ± *SD*)
	EX3	x	x	(x)	(x)	x	(x)	x	x	(x)	x	(x)	x	7	c: undergraduate students; d: no information regarding participants’ gender and health condition (only vision was reported); h: only a reference is made to EX1; k: no report of *SD*, CI, or variation of any kind for the congruent and incongruent stimuli; m: no report of the Stroop effect (*M* ± *SD*)
Schwartz et al. (2018) [16]		x	x	(x)	x	x	(x)	x	x	(x)	x	(x)	x	8	c: undergraduate students; h: no description of when and how the RTs were extracted specifically; k: no report of the *M* and *SD* for congruent and incongruent stimuli (the data from the alternating condition was requested separately for sitting and standing); m: no report of the Stroop effect (*M* ± *SD*)
Zhang et al. (2018) [36]		x	x	(x)	x	x	(x)	(x)	x	x	x	(x)	x	9	c: undergraduate students; h: no description of when and how the RTs were extracted specifically; i: small sample size; m: no report of the Stroop effect (*M* ± *SD*)
Smith et al. (2019) [13]		x	x	(x)	(x)	x	x	(x)	x	x	x	(x)	x	8	c: undergraduate students; d: no information regarding participants’ gender and health condition; i: small sample size, although they reference Rosenbaum et al. (2017) saying that the sample size is similar; m: no report of the Stroop effect (*M* ± *SD*)
Caron et al. (2020) [29]	EX1	x	x	(x)	(x)	x	(x)	x	x	x	x	(x)	x	8	c: undergraduate students; d: no information regarding participants’ gender and health condition (only vision was reported); h: no description of when and how the RTs were extracted specifically; m: no report of the *SD* of the Stroop effect (*M* ± *SD*)
	EX2	x	x	(x)	(x)	x	(x)	x	x	x	x	(x)	x	8	c: undergraduate students; d: no information regarding participants’ gender and health condition (only vision was reported); h: no description of when and how the RTs were extracted specifically; m: no report of the *SD* of the Stroop effect (*M* ± *SD*)
	EX3	x	x	(x)	(x)	x	(x)	x	x	x	x	(x)	x	8	c: undergraduate students; d: no information regarding participants’ gender and health condition (only vision was reported); h: no description of when and how the RTs were extracted specifically; m: no report of the *SD* of the Stroop effect (*M* ± *SD*)
	EX5	x	x	(x)	(x)	x	(x)	x	x	x	x	(x)	x	8	c: undergraduate students; d: no information regarding participants’ gender and health condition (only vision was reported); h: no description of when and how the RTs were extracted specifically; m: no report of the *SD* of the Stroop effect (*M* ± *SD*)
Straub et al. (2022) [30]	EX1	x	x	(x)	(x)	x	(x)	x	x	x	x	(x)	x	8	c: undergraduate students; d: no information regarding standard deviation in age and health condition; h: no description of when and how the RTs were extracted specifically; m: no report of the Stroop effect (*M* ± *SD*)
	EX2	x	x	(x)	(x)	x	(x)	x	x	x	x	(x)	x	8	c: undergraduate students; d: no information regarding standard deviation in age and health condition; h: no description of when and how the RTs were extracted specifically; m: no report of the Stroop effect (*M* ± *SD*)
Pinho et al. (2022) [37]		x	x	(x)	x	x	(x)	(x)	x	x	x	x	x	9	c: only women were included in this study and in the group of younger women there are only very young women, presumably students as the study was performed in the university setting; h: no description of when and how the RTs were extracted specifically; i: small sample size

Note. x = yes; (x) = was partially done, general remarks. a, sufficient description of question/objective; b, appropriate study design; c, appropriate method of participant selection or source of information/input variables; d, sufficient description of patient characteristics; e, description of interventional and random allocation; h, report of means of assessment with outcome measures well-defined and robust to measurement/misclassification bias; i, appropriate sample size; j, appropriate analytic methods, and method description; k, report of the estimate of variance in main results; l, control for confounding; m, sufficiently detailed report of results; n, conclusions supported by the results.

**Table 2 ijerph-20-02319-t002:** Characteristics of included studies.

Experiments within Study	Study Aims	Participants (*N*, Age (*M*), Age (*SD*/Range), Sex)	Measures and Study Design	Exposure Duration	Outcome	Findings
Rosenbaum et al. (2017): EX1 [28]	Effect of sitting vs. standing on Stroop performance	*N* = 17, *M* = 23 y, age range = 19–27 y, n.a.	In the Stroop task, the stimuli were the color words “RED”, “GREEN”, “BLUE”, “BROWN” combined factorially with the corresponding print colors. The stimuli were generated in Microsoft Word and displayed on a light gray background on a 14-inch color monitor. Viewed from a distance of approximately 60 cm, participants responded by speaking into a microphone the name of the print color in which the words appeared. During both conditions, sitting and standing, participants were presented with 72 color-word Stroop stimuli, half congruent, half incongruent. The order of testing between sitting and standing was counterbalanced randomly across participants.	n.a.	RT, PE	Decrease in the Stroop effect when participants were standing.
Rosenbaum et al. (2017): EX3 [28]		*N* = 50, *M* = 26.1 y, age range = 19–32 y, n.a.	The stimuli and design were the same as in EX1.	n.a.		Decrease in the Stroop effect when participants were standing; ruled out the absolute RT as the factor generating the difference in selectivity between the standing and sitting conditions.
Schwartz et al. (2018) [16]	Effect of alternating postures on cognitive performance for healthy people performing sedentary work	*N* = 30, *M* = 25.3 y, *SD* = 3.8 y, 14 women	A digital Stroop test containing 190 congruent, incongruent and neutral (i.e., four crosses “XXXX” written in different colors) items were used. Text editing task and d2R test were also used. Reaction time and working speed were measured and recorded automatically using the software. For the intervention arm, the battery blocks were executed in alternating postures (sit-stand-sit-stand-sit) and for control periods (non-intervention day), all five battery trials were conducted in a sitting posture (sit-sit-sit-sit-sit); cross-over design.	8–10 min (the whole block app. 30 min); a fixed 5-min break occurred between successive batteries	RT	No significant difference in the Stroop task between standing and sitting trials was found.
Zhang et al. (2018) [36]	Impact of the use of active workstation on executive function	*N* = 24, 12 men: *M* = 24 y, *SD* = 1.5 y, 12 women: *M* = 22.1 y, *SD* = 1.5 y	In the Stroop task, there were six kinds of trials: 1) the word “red” printed in red ink, 2) the word “blue” printed in blue ink, and 3) the word “green” printed in green ink, which was regarded as a congruent condition, and 4) the word “red” printed in blue or green ink, 5) the word “blue” printed in red or green ink, and 6) the word “green” printed in red or blue ink, which was regarded as an incongruent condition. Each stimulus was presented at 2000 ms and between two stimuli was 2 to 8 s interval with the sign “+” presented on the screen. Participants were required to tell the color name of the words by pressing corresponding buttons on the keyboard. N-back test and more-odd shifting task were also used. There were 96 color-word Stroop stimuli, half congruent and half incongruent. Each participant performed a test battery under each of four conditions, including sitting, standing, walking at an active workstation with self-selected speed (self-paced walking), and walking at an active workstation with 1.5 times the self-selected speed (faster walking). The order in which participants performed the experimental conditions was counterbalanced.	n.a. (the whole block app. 25 min); there were 2 min intervals between each task	RT, PE	The Stroop task performance did not vary across four workstation conditions.
Smith et al. (2019) [13]	Replication of the findings of Rosenbaum et al. (2017)	*N* = 14, undergraduate students, n.a.	In the Stroop task, there were three kinds of letter strings: 1) congruent strings spelled color-words that were consistent with the color in which they appeared (e.g., “RED” appearing in red), 2) incongruent strings spelled color-words that were inconsistent with the color in which they appeared, but consistent with the alternative response (e.g., “RED” appearing in green), 3) neutral strings consisted of a series of Xs (three or five Xs long to match the lengths of the strings “RED” and “GREEN”) in red or green. For all three kinds of letter strings, green and red strings were presented equally often. All stimuli were presented on an LCD flat-panel display on a black background. There was a 1500 ms inter-trial interval with a white fixation cross presented at the center of the screen in-between trials. Participants indicated their response by pressing the corresponding response button and they heard an error tone if they pressed the wrong button or did not respond within 1500 ms. Participants performed one-half of the experiment in each posture, sitting and standing, with posture order counterbalanced across subjects. For each posture, there were two initial blocks of practice trials, followed by four blocks of experimental trials. Each block (practice and experimental) consisted of 12 neutral, 12 congruent, and 12 incongruent trials for a total of 144 experimental trials in each posture. There was a brief break between blocks.	30 min	RT, PE	The magnitude of Stroop interference was markedly reduced when participants adopted a standing posture; there was no main effect of posture, suggesting that the mild postural control requirements associated with standing enhanced attentional selectivity with no apparent cost.
Caron et al. (2020): EX1 [29]	Replication and extension of the findings of Rosenbaum et al. (2017)	*N* = 107, undergraduate students, n.a.	The Stroop task consisted of congruent stimuli (the uppercase words “RED”, “GREEN”, “BLUE”, and “BROWN”, each presented in their matching hue), incongruent stimuli (included all other combinations of the color words and hues), and neutral stimuli (consisted of colored strings of three to five Xs matched to the number of letters in the four-color words). The stimuli were displayed on a desktop PC on a light gray background. On each trial, the letter string appeared for 2000 ms, after which it was replaced by a gray screen for 2000 ms. Vocal responses were collected using a noise-canceling microphone. The experiment consisted of one block of 48 practice trials followed by two blocks of 108 experimental trials (36 congruent, 37 incongruent, 36 neutral, with 9 repetitions of each hue). One experimental block was performed sitting and the other standing. The order of posture conditions was counterbalanced across participants.	n.a.	RT, PE	The Stroop effect was smaller when participants were standing than when sitting but only for participants who started in the sitting posture.
Caron et al. (2020): EX2 [29]		*N* = 107, undergraduate students, n.a.	The stimuli and design were the same as in EX1, with three exceptions: (1) there were 24 practice trials before the initiation of both the sitting and standing condition, (2) there were only congruent and incongruent trials (no neutral trials), and (3) the number of trials in each condition was increased to 60.	n.a.		The Stroop effect was smaller when participants were standing than when sitting but only for participants who started in the sitting posture.
Caron et al. (2020): EX3 [29]		*N* = 97, undergraduate students, n.a.	The Stroop task consisted of color words “RED”, “YELLOW”, “BLUE”, and “GREEN” in colors for red, yellow, blue, and green, respectively. The stimuli were displayed on the E-Prime. Each trial began with a fixation cross in the center of the screen for 500 ms followed by a blank screen for 500 ms. The target display was then presented and remained on the screen until a response was made. Feedback was displayed when an error was made. Feedback was provided on incorrect trials. Responses were collected with a keyboard held vertically to the chest in both conditions. The experiment consisted of two blocks of 88 color-word Stroop stimuli, half congruent, half incongruent. One experimental block was performed sitting and the other standing. The order of posture conditions was counterbalanced across participants.	n.a.		The Stroop effect was smaller when participants were standing than when sitting but only for participants who started in the sitting posture.
Caron et al. (2020): EX5 [29]		*N* = 50, undergraduate students, n.a.	The stimuli and design were the same as in EX1, with four exceptions: (1) there were no practice trials, (2) the number of trials in each condition was increased to 72, (3) on each trial, the letter string appeared and remained on the screen until a response was made, after which it was replaced by a gray screen for 1000 ms before the stimulus for the next trial was presented, and (4) vocal responses were collected using a voice key (connected to a microphone).	n.a.		Responses for a given posture were faster when they were in Block 2 than in Block 1 (see Supplementary Material of the study [29]). https://osf.io/8h52v/
Straub et al. (2022): EX1 [30]	Replication of the findings of Rosenbaum et al. EX1 (2017)	*N* = 44, *M* = 25.8 y, *SD* = n.a., 30 women	The Stroop task consisted of four different colors (green, red, blue, yellow) in which the carrier words (“GRÜN”, “ROT”, “BLAU”, and “GELB”, respectively) written in German in capital letters were presented against a black background. Stimuli were either congruent (e.g., “GRÜN” written in green) or incongruent (e.g., “GRÜN” written in yellow). Congruency was manipulated trial-wise with an equal proportion of congruent and incongruent stimuli. The stimuli were displayed on the E-Prime. A trial consisted of the presentation of a target stimulus that remained on the screen for 2000 ms or until a response was registered, followed by an inter-trial interval of 500 ms. Responses were collected manually. Error feedback was presented (for 500 ms) when participants pressed the wrong key or did not respond within the given response window. Before the experiment started, participants completed a training session with 20 trials in the posture they were assigned to start with. Participants completed one block with 144 trials in one posture and then changed the posture to complete another block in the other posture. The experiment consisted of four blocks with 576 trials in total. The order of posture was counterbalanced across participants.	n.a.	RT, PE	Did not confirm the hypothesis that the Stroop effect differs between standing and sitting posture.
Straub et al. (2022): EX2 [30]	Replication of the findings of Rosenbaum et al. EX1 (2017)	*N* = 38, *M* = 23.47 y, *SD* = n.a., 30 women	Each trial started with a fixation cross presented on the screen for 500 ms, followed by a color word (i.e., “GRÜN” or “ROT”) or neutral letters that consisted of a series of three or five Xs matched to the number of letters in the two color words presented in either green or red. Congruency was manipulated trial-wise with an equal proportion of congruent and incongruent stimuli. There was a 1500 ms inter-trial interval before the next fixation cross occurred. Responses were collected manually. An error tone was presented if participants pressed the wrong key or did not respond within 1500 ms. Each condition of posture started with two training blocks followed by four blocks of experimental trials. Each block comprised 12 neutral, 12 congruent, and 12 incongruent stimuli resulting in 36 trials per block and a total of 144 experimental trials per posture. The order of posture was counterbalanced across participants.	n.a.	RT, PE	Did not confirm the hypothesis that the Stroop effect differs between standing and sitting posture.
Pinho et al. (2022) [37]	To study the effects of a sensory aid (a light touch) on a dual-task paradigm and to understand the different responses on balance that were due to aging.	*N* = 25 women, *M* = 24.2 y, *SD* = 4.0 y, (also 25 older women were included (*M* = 67.3 y, *SD* = 4.2 y))	Only incongruent stimuli were used (color and word font differ). Four different colors were used (red, blue, black, and green) and twelve sequences of six capitalized letters words were previously established in a randomized and balanced order (one for each condition with the Stroop test and three extras if it was necessary to repeat a trial). The sequence began with an alert symbol (+) indicating that the test would begin in 10 s. Then, the word appeared and remained in the projection for 3 s each. Each time a new word was projected, a beep was played to synchronize the participant’s answer and the stimuli onset. A microphone was used to register the participants` performance. Participants performed in a random and balanced order three trials of each condition (seated, quiet standing balance, quiet standing balance with haptic input via light touch, quiet standing balance with Stroop test, quiet standing balance with Stroop test and with haptic input via light touch) with a rest interval of 60 s between trials.	n.a.	RT, PE	Young adults showed reduced reaction time than older adults. The reaction time in correct answers did not differ between conditions. No differences between conditions were found for the reaction time of wrong answers.

Note. EX = experiment, RT = reaction time, PE = percentage of error; n.a. = not available.

## Data Availability

The datasets generated and analyzed during this meta-analysis are available from the corresponding author on request.

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
