# Peer review of "Does Standing Up Enhance Performance on the Stroop Task in Healthy Young Adults? A Systematic Review and Meta-Analysis"

_ijerph, 2023, doi:10.3390/ijerph20032319_

Round 1
Reviewer 1 Report
This paper reviewed and examined the Stroop effect between standing and sitting. There are several areas of this paper that require further improvement.
INTRODUCTION
1) Given the authors' focus on the Stroop effect, there should be a more comprehensive explanation of the Stroop effect in the introduction as well as a comparison with other cognitive tasks, such as N-back, Simon, Flanker, and Go/NoGo. How are they different/ similar, and the rationale to focus on the Stroop effect only?
METHODS
2) Why was the review limited to randomised controlled trials? For a comprehensive systematic review, studies adopting other forms of experimental designs should also be considered. In addition, the grey literature should also be considered as they also make important contributions (e.g., publication bias) to a systematic review.
3) Table 1 could be better organised as opposed to placing all information in the remark column.
RESULTS
4) The 95% confidence interval values reported in-text regarding the Stroop effect, congruent trials, and incongruent trials were inconsistent with the values reported in Figures 2, 3, and 4. For example, line 260 refers to the Stroop effect but Figure 2 indicates congruent effects. This makes the meta-analytic findings and associated discussion points in the subsequent section difficult to interpret.
5) As mentioned by the authors, there is significant heterogeneity across studies. The meta-analysis should take into account variations across the Stroop tasks. For instance, the authors should report the duration of the stimuli presented, intertrial interval, the total number of trials, and the number of trials within a block (if the task was conducted in blocks) in Table 2.
6) In addition, the authors should also clearly indicate the number of trials and the number of unique stimuli separately in Table 2. For example, the authors reported "144 color-word Stroop stimuli", how many unique stimuli were there?
DISCUSSION
7) Given the confusing result section, the discussion section could not be adequately evaluated.
Author Response
INTRODUCTION
- Given the authors' focus on the Stroop effect, there should be a more comprehensive explanation of the Stroop effect in the introduction as well as a comparison with other cognitive tasks, such as N-back, Simon, Flanker, and Go/NoGo. How are they different/ similar, and the rationale to focus on the Stroop effect only?
Response 1.1: Thank you for this comment. We agree that the Stroop effect should be better explained and also compared to other cognitive tasks that are generally used to assess inhibitory control. To address this comment, we added another paragraph to the introduction (cf. lines 78-106). The main focus of this manuscript was on the Stroop effect as this was the most commonly used measure in the reviewed literature concerning differences in cognitive performance during sitting and standing. Furthermore, based on contradictive results regarding differences in the Stroop effect during sitting and standing obtained by Rosenbaum et al. (2017), Smith et al. (2019), Caron et al. (2020), and Straub et al. (2022), we wanted to perform a meta-analysis solely for this cognitive measure.
METHODS
- Why was the review limited to randomised controlled trials? For a comprehensive systematic review, studies adopting other forms of experimental designs should also be considered. In addition, the grey literature should also be considered as they also make important contributions (e.g., publication bias) to a systematic review.
Response 1.2: Thank you. This was a mistake that was now corrected (cf. line 149 and 163). The review was limited to crossover randomised controlled trials as participants had to receive both interventions (i.e., performing the Stroop task during sitting and standing), but the order in which they received interventions was randomised. For example, if the design of the study only included performing the Stroop task during sitting/standing, we did not consider it suitable for this meta-analysis. Likewise, if the Stroop task was performed in the alternating condition, we only considered it if data for sitting and standing separately were available or provided by the authors. This decision was based on the fact that there are many variants of Stroop that involve differences in response times. Already the original work by Rosenbaum et al. (2017) showed that there is an effect, but it is small. For this reason, a crossover randomised controlled trials seem to be the most appropriate designs.
- Table 1 could be better organised as opposed to placing all information in the remark column.
Response 1.3: In the remark column of Table 1 we reported quality assessment criteria as defined by the Standard Quality Assessment Criteria (SQAC) that we used. We described these criteria in detail in the “Selection Criteria” section and only noted their meaning in the remark column to make it quickly accessible when reading the Table 1. Did you perhaps mean table 2 where we summarized all the outcomes? We believe that with Tables 1 and 2, the reader can get a comprehensive overview of the quality of the included studies, the design, and the main results with outcomes. If you feel that this could be improved, please make a suggestion or give us an example so that we can restructure the table. We thank you for your support.
RESULTS
- The 95% confidence interval values reported in-text regarding the Stroop effect, congruent trials, and incongruent trials were inconsistent with the values reported in Figures 2, 3, and 4. For example, line 260 refers to the Stroop effect but Figure 2 indicates congruent effects. This makes the meta-analytic findings and associated discussion points in the subsequent section difficult to interpret.
Response 1.4: Thank you very much for this comment. Indeed, this was a mistake (Figures were not properly addressed and placed) that was now corrected (cf. “Meta-Analysis Outcomes” section).
- As mentioned by the authors, there is significant heterogeneity across studies. The meta-analysis should take into account variations across the Stroop tasks. For instance, the authors should report the duration of the stimuli presented, intertrial interval, the total number of trials, and the number of trials within a block (if the task was conducted in blocks) in Table 2.
Response 1.5: Thank you for this comment. We agree with you that we did not report enough data in Table 2. Measures and study design have now been redone to include all data that was reported by the authors.
- In addition, the authors should also clearly indicate the number of trials and the number of unique stimuli separately in Table 2. For example, the authors reported "144 color-word Stroop stimuli", how many unique stimuli were there?
Response 1.6: Thank you for this comment. We agree with you that we did not report enough data in Table 2. Measures and study design have now been redone to include all data that was reported by the authors.
DISCUSSION
- Given the confusing result section, the discussion section could not be adequately evaluated.
Response 1.7: We feel that with suggestions of three reviewers and editor our manuscript significantly improved. Taking into account all suggestions we upgraded also the discussion section. Thank you very much once again.
Reviewer 2 Report
The authors reviewed randomized controlled trials on the effect of sitting and standing on Stroop performance and performed meta-analysis to compare how standing compared to sitting affected Stroop performance in each of the conditions. The study addresses conflicting findings in the literature regarding the effect of standing on a task utilizing executive control. Careful analyses by the authors show lack of significant differences between sitting and standing for the Stroop effect, although a nonsignificant trend was found showing better cognitive processing in standing compared to sitting.
A recent study by Straub et al. (2022) on the same subject conducted two replications of the studies from Rosenbaum et al. (2018) (experiments 1 & 3) and Smith et al. (2019), which are included in the meta-analysis presented in the manuscript, and a meta-analysis of three of the five studies that the authors included in their meta-analysis. (The authors cite Rosenbaum et al. 2017 whereas Straub et al. cite Rosenbaum et al. 2018 but they appear to be the same studies). Given the similarities of the Straub et al. study with the present study and the similar results it would be important to incorporate this study in the Discussion and stress what their study contributes to the literature.
Minor comments:
The authors include the age range of 18-65 years as one of the selection criteria (p. 3 l. 128), yet the studies selected all involve college undergraduates. Does this mean that no studies of older persons were found to meet criteria? On p. 7 (l. 214) of the 37 potential publications that were identified five met the inclusion criteria and one study was excluded because of age. Does this mean that participants of the excluded study were older or younger than the age range of the criteria?
On p. 4 l. 166 the authors introduce the five studies on which information was obtained before explaining how these five studies were derived from the publications reviewed. This information is provided further down, on p. 7. A re-phrasing on the sentence on l. 166 and/or a link to the information provided further down would help the reader make the transition.
On p. 8 it would help the reader if the 9 experiments were explicitly associated with the 5 publications reviewed, perhaps on l. 225; this is not immediately obvious, and the reader might search for 9 separate publications.
The phrasing on p. 13 l. 268-270, “ranging from non-significantly small … to non-significantly small effect size…” is confusing and should be edited to express the meaning more clearly. The same applies to the phrasing further down, l. 281-282.
Author Response
The authors reviewed randomized controlled trials on the effect of sitting and standing on Stroop performance and performed meta-analysis to compare how standing compared to sitting affected Stroop performance in each of the conditions. The study addresses conflicting findings in the literature regarding the effect of standing on a task utilizing executive control. Careful analyses by the authors show lack of significant differences between sitting and standing for the Stroop effect, although a nonsignificant trend was found showing better cognitive processing in standing compared to sitting.
- A recent study by Straub et al. (2022) on the same subject conducted two replications of the studies from Rosenbaum et al. (2018) (experiments 1 & 3) and Smith et al. (2019), which are included in the meta-analysis presented in the manuscript, and a meta-analysis of three of the five studies that the authors included in their meta-analysis. (The authors cite Rosenbaum et al. 2017 whereas Straub et al. cite Rosenbaum et al. 2018 but they appear to be the same studies). Given the similarities of the Straub et al. study with the present study and the similar results it would be important to incorporate this study in the Discussion and stress what their study contributes to the literature.
Response 2.1: We are very thankful for this comment. We found the study of Straub et al. (2022) right after the first submission of this manuscript. We now incorporated it in the introduction and discussion. Moreover, taking into account suggestions from other reviewers and editor, we reran the systematic search and included two more studies.
Minor comments:
- The authors include the age range of 18-65 years as one of the selection criteria (p. 3 l. 128), yet the studies selected all involve college undergraduates. Does this mean that no studies of older persons were found to meet criteria? On p. 7 (l. 214) of the 37 potential publications that were identified five met the inclusion criteria and one study was excluded because of age. Does this mean that participants of the excluded study were older or younger than the age range of the criteria?
Response 2.2: Thank you for these questions. Yes, that’s right, no studies of older persons were found that would meet the criteria and all included studies were performed on undergraduate students. And as for the second question, participants of the excluded study were older - M = 71, SD = 4 (Feasibility and behavioral effects of prolonged static and dynamic standing as compared to sitting in older adults with type 2 diabetes mellitus (Marusic et al., 2020)). When the second search was performed after the first reviews came in, we found altogether 18 studies that were excluded due to participants’ age (these studies were either performed on children/adolescents or older people).
- On p. 4 l. 166 the authors introduce the five studies on which information was obtained before explaining how these five studies were derived from the publications reviewed. This information is provided further down, on p. 7. A re-phrasing on the sentence on l. 166 and/or a link to the information provided further down would help the reader make the transition.
Response 2.3: Thank you for this comment. We now added citations of the five included studies and a link to the information on how studies were derived (cf. lines 211-214).
- On p. 8 it would help the reader if the 9 experiments were explicitly associated with the 5 publications reviewed, perhaps on l. 225; this is not immediately obvious, and the reader might search for 9 separate publications.
Response 2.4: Thank you for this comment. We realize we mention nine experiments for the first time in the text here and it can come as a surprise, so we added citations of the nine included experiments.
- The phrasing on p. 13 l. 268-270, “ranging from non-significantly small … to non-significantly small effect size…” is confusing and should be edited to express the meaning more clearly. The same applies to the phrasing further down, l. 281-282.
Response 2.5: We agree that the phrasing was difficult to understand. We now rephrased it throughout the text in each subsection (Stroop effect, congruent and incongruent analyses). With that we would like to thank you once again for all your great suggestions.
Reviewer 3 Report
I reviewed the manuscript entitled " Does Standing Up Enhance Performance on the Stroop Task in Healthy Young Adults? A Systematic Review and Meta-Analysis". There are some remarks which should be considered:
1- Search databases are not enough to call the study a systematic review. Medline and PubMed have a vast overlap, it is recommended to do the search through datasets like WOS, Embase, etc. More studies could be found in this way.
2- The search date belongs to more than 1 year ago, so it is recommended to update the search.
3- Why did the authors use Hedge's g as their effect size? In comparison with Cohen's d.
4- Which statistical software was used?
5- The importance of the study is not well-described, especially, the clinical significance.
6- The conclusion could not be concluded by combining only five heterogeneous studies. It should be stated with more caution.
7- PRISMA is not accurate, the qualitative synthesis doesn't mean meta-analysis. All of the studies that have been entered in qualitative synthesis should be reported in Table-a, besides, the risk of bias assessment should be done on them.
Author Response
I reviewed the manuscript entitled " Does Standing Up Enhance Performance on the Stroop Task in Healthy Young Adults? A Systematic Review and Meta-Analysis". There are some remarks which should be considered:
- Search databases are not enough to call the study a systematic review. Medline and PubMed have a vast overlap, it is recommended to do the search through datasets like WOS, Embase, etc. More studies could be found in this way.
Response 3.1: Thank you for this comment. In line with your second comment, we decided to do another search, this time through the following databases: Academic Search Complete, CINAHL Ultimate, MEDLINE, PubMed, and Web of Science. Unfortunately, we do not have permission to use the Embase through our institutional accounts. Two new and suitable studies that met the inclusion criteria were found. Straub et al. (2022) performed two experiments as a replication of the studies by Rosenbaum et al. (2017) as well as a meta-analysis on whether body posture reduces the Stroop effect. The second study that met the inclusion criteria was by Pinho et al. (2022) in which the authors investigated the effects of a sensory aid (a light touch) on a dual-task paradigm (cf. Table 2 on line 306). Since in both studies raw data was requested and not yet received, it was impossible to calculate the Stroop effect (especially the standard deviations). For these reasons we updated all the results but for Stroop effect. Although we added three more experiments (e.g., for incongruent stimuli 104 new subjects), the results did not change and still point to the same direction.
- The search date belongs to more than 1 year ago, so it is recommended to update the search.
Response 3.2: We agree with you. As mentioned above, we did another systematic search and all studies that were published until the 31st of December 2022 were considered this time.
- Why did the authors use Hedge's g as their effect size? In comparison with Cohen's d.
Response 3.3: Thank you for this question. In our view, Hedge's g is generally preferable to Cohen's d statistic because it has better properties for small samples and better properties when sample sizes are significantly different. In our software we could choose Standard difference in means, Difference in means, Hedge’s g and Standard paired difference. For instance, for the incongruent stimuli, the Standard difference in means shows the effect of 0.18 (p=0.165) in comparison with Hedge’s g, which we reported and is 0.18 (p=0.164). Similar for both Stroop effect and congruent stimuli, the effect remains very similar.
- Which statistical software was used?
Response 3.4: Analyses were performed using the program Comprehensive Meta-Analysis (CMA version 2, Biostat, Englewood, NJ, USA). We mentioned this in lines 225, thank you.
- The importance of the study is not well-described, especially, the clinical significance.
Response 3.5: Thank you for this suggestion. So far, there is not much available on clinical population. For instance, our paper is currently under review (Marusic et al., in revision) and describes the effects of prolonged standing in patients with Parkinson’s disease. We now described the clinical importance of the study a bit more (cf. lines 473-482). The text goes along these lines: “Despite the known health benefits of physical activity, older adults generally remain sedentary for an average of 9.4 hours per day (up to 80% of their waking day) (Harvey, Chastin, & Skelton, 2015). These numbers are even higher for frail populations such as diabetics and those with Parkinson's disease, suggesting that the need for physical activity is not being met among older adults who need it most. Given the known barriers to physical activity participation in older patients with chronic conditions, a natural first step would be to reduce physical inactivity through intermittent non-exercise physical activity such as standing. This raises the question of neurocognitive performance, which is currently poorly studied in such populations.”
- The conclusion could not be concluded by combining only five heterogeneous studies. It should be stated with more caution.
Response 3.6: Thank you for this suggestion. We agree that our meta-analysis only included seven studies / twelve experiments that have heterogeneous study designs and measures and more research is needed to make more valid conclusions. We now changed this part a bit (cf. lines 495-496).
- PRISMA is not accurate, the qualitative synthesis doesn't mean meta-analysis. All of the studies that have been entered in qualitative synthesis should be reported in Table-a, besides, the risk of bias assessment should be done on them.
Response 3.7: Thank you. Figure 1 was updated after the second search was performed. We also changed the phrasing in the lowest square of this Figure so that it says “Studies included in meta-analysis”. All of the studies included in the meta-analysis are then mentioned in the following text (cf. “Characteristics of Included Studies” section). Once again, we would like to sincerely thank you for all your provided support.
Round 2
Reviewer 2 Report
Response 2.1. The authors have incorporated the Straub et al. (2022) study in the Introduction and have included three additional studies (two according to their response). In the revised manuscript (p. 3 l. 359), given the similarities of their study and Straub et al., they stress as their contribution to the literature is that Straub et al. did not investigate “whether standing compared to sitting affects the reaction time for congruent and incongruent stimuli separately”. The Straub et al. study, however, analyzed reaction time with congruency (congruent vs incongruent) as within-subject factors. It is therefore still unclear what their study contributes to the literature given their results are similar.
Response 2.2. Given that all studies were performed on undergraduates, this should be stressed in the paper. The study is essentially a review and meta-analysis of enhancement in undergraduates. As expressed, the age range as a selection criterion is misleading. Moreover, the age range selected is extremely wide and theoretical justification is needed on why the upper limit was 65 and not 70, for example. Given that younger people typically have good attention and executive functions, it is conceivable that the effect of standing vs sitting might be more salient in an older age group. How many of the excluded studies had older participants and what were the mean ages of them?
Response 2.4. I am not sure where you have added the citations of the 9 experiments. Can you provide the line numbers?
Author Response
The authors have incorporated the Straub et al. (2022) study in the Introduction and have included three additional studies (two according to their response). In the revised manuscript (p. 3 l. 359), given the similarities of their study and Straub et al., they stress as their contribution to the literature is that Straub et al. did not investigate “whether standing compared to sitting affects the reaction time for congruent and incongruent stimuli separately”. The Straub et al. study, however, analyzed reaction time with congruency (congruent vs incongruent) as within-subject factors. It is therefore still unclear what their study contributes to the literature given their results are similar.
Response 2.1.: Dear reviewer #2, thank you once again for your effort and constructive comments. Indeed, we incorporated two studies and three experiments as Straub et al. (2022) performed two experiments within one study. In the two experiments of Straub et al that were conceptual replications of the study of Rosenbaum et al. (2017), “reaction times and errors were analyzed with a frequentist repeated measures ANOVA with congruency (congruent vs incongruent) and posture (sitting vs standing) as within-subject factors”. However, Straub et al. performed meta-analysis only for the Stroop effect and not also separately for congruent and incongruent stimuli. In other words, in their experiments, they recorded reaction times for congruent and incongruent stimuli separately both in sitting and standing condition but performed meta-analysis only for the Stroop effect in general. What our study contributes to the literature is that we included three additional studies in our meta-analysis (Schwartz et al. (2018), Zhang et al. (2018), Pinho et al. (2022)) and performed a meta-analysis not only for the Stroop effect but also for congruent and incongruent stimuli separately (cf. lines 134-137). With a completely new systematic search (performed until December 31, 2022) and the included studies, we believe that our manuscript adds a clear statement to the literature that no significant difference was found between sitting and standing for the Stroop effect, even when congruent and incongruent stimuli are compared separately.
Given that all studies were performed on undergraduates, this should be stressed in the paper. The study is essentially a review and meta-analysis of enhancement in undergraduates. As expressed, the age range as a selection criterion is misleading. Moreover, the age range selected is extremely wide and theoretical justification is needed on why the upper limit was 65 and not 70, for example. Given that younger people typically have good attention and executive functions, it is conceivable that the effect of standing vs sitting might be more salient in an older age group. How many of the excluded studies had older participants and what were the mean ages of them?
Response 2.2.: Thank you for sharing your thought on this. Our initial idea was to perform a meta-analysis and look into differences in posture when performing the Stroop task in sitting and standing for healthy adults. As developmental psychology theoretically marks adulthood between 18 and 65 years of age, we decided to consider all studies that performed the Stroop task in sitting and standing on participants between 18 and 65 years old. Also, the majority of studies that were excluded due to participants’ age (n = 18, cf. flow diagram) performed the Stroop task on participants older than 65. However, it is true that, for example, a reported mean age of 70 years may include participants younger than 65 years (this was our inclusion/exclusion criterion). An article was therefore excluded from further reading if the age range was > 65 years. Considering that your observations are accurate and that our meta-analysis refers to young adults ( undergraduates), we reported/added a number of things:
- In the title we use “in healthy young adults”
- Stressed in the abstract “We pooled data from a total of 603 healthy young adults…”
- We added text to the abstract to further highlight this fact: “Importantly, these results imply that changing from a seated to a standing posture in healthy young adults is unlikely to have detrimental effects on selective attention and cognitive control. To gain a full understanding of this phenomenon, further research should examine this effect in a population of healthy older adults as well as in a population with pathology.”
- Row 190: “When the sample consisted of undergraduate students but results were generalized across healthy adults of different ages a general remark was noted”.
- First paragraph in Discussion highlighting that this no significant effect was found in healthy young adults.
- In discussion from row 419 to 446 we discuss why this is the case in young adults and could be different in older or those with (neurodegenerative) pathologies.
- We added to the limitation section where we highlighted part of your comment: “Moreover, because these effects may be more pronounced in an older age group, further research should examine the effects of standing versus sitting on selective attention and cognitive control in both a population of healthy older adults and a population with pathology”.
- In conclusions we point to this fact: “Future research is needed to support these conclusions and should focus on assessing cognitive-motor interference in healthy older adults and pathological populations, as well as in more complex postural conditions and other executive functions. Nevertheless, caution should be exercised in interpreting these results, as the robustness of the analysis and generalization is limited.”
I am not sure where you have added the citations of the 9 experiments. Can you provide the line numbers?
Response 2.4.: Thank you for this comment. Indeed, this information must have slipped out of the previous text. Citations of these experiments are added in lines 282-284.
Thank you again, and for your information, we have additionally proofread the entire text and all changes have change marks (track-changes).
Reviewer 3 Report
My comments have been addressed, except for the English that needs revision.
Author Response
My comments have been addressed, except for the English that needs revision.
Response: Thank you very much. English was now revised. The text was sent to proofreading.